# Towards Sustainability: New Tools for Planning Urban Pedestrian Mobility

**Daniela Santilli** [1,*], **Mauro D'Apuzzo** [1], **Azzurra Evangelisti** [1] **and Vittorio Nicolosi** [2]

1   Department of Civil and Mechanical Engineering DICEM, University of Cassino and Southern Lazio, 03043 Cassino, Italy; dapuzzo@unicas.it (M.D.); aevangelisti.ing@gmail.com (A.E.)
2   Department of Enterprise Engineering "Mario Lucertini", University of Rome "Tor Vergata", 00133 Rome, Italy; nicolosi@uniroma2.it
*   Correspondence: daniela.santilli@unicas.com

**Abstract:** Background: Since the beginning of the new millennium, sensitivity towards the environment has been spreading globally. In fact, countries are adopting measures to develop new decision support tools that can evaluate the impact of interventions to promote and encourage sustainable mobility. To reduce the levels of pollution related to road traffic, policies that favor multimodal transport alternatives have been strengthened. This involves the combined use of public transport, cycling and walking paths, as well as sharing services where available. Regardless of the type of transport, the pedestrian component remains relevant in cities, even if the infrastructures are often not adequate to accommodate it and conflicts arise that must be managed. It is, therefore, necessary to assess the exposure to risk in terms of road safety. Methods: To this end, the work proposes a forecasting model to estimate the pedestrian flows that load the network. The methodology employs a hybrid approach that appears to better capture the movements of pedestrians. Results: By comparing the results of the model with the real data collected on the study area, satisfactory estimates were obtained. Conclusions: Therefore, this can be an effective tool to help road managers to evaluate the actions to protect vulnerable users.

**Keywords:** soft mobility; pedestrian flows; space syntax; hybrid model

## 1. Introduction

The quality of life in cities in recent decades has deteriorated due to the growing need for the movement of people and goods, which has unexpectedly increased during these long months of the pandemic. In fact, together with the birth and diffusion of new information technologies, mobility has changed the life of individuals, allowing them to move to farther areas in shorter times. This has led to a radical change in the socio-economic and territorial structure of the city, which has seen a greater diffusion of motorized means of transport allowing the development of productive activities in areas separated from residential areas. So, workers have begun to move systematically, that is to move at regular times and along the fixed routes that connect the neighborhood of residence to production and industrial areas.

In this new type of city, services and activities are no longer located exclusively close to residences and in the central areas but have spread to the sub- and peri-urban areas, giving rise to a polycentric or "patchy" urban development. Precisely due to the dispersion of services over large areas and the need to interconnect the various areas related to different aspects of social life, users have become dependent on private cars [1,2].

The advent of the automobile has meant that urban trips are made by motor vehicles to the detriment of other modes (such as walking, cycling or public transport, etc.) which have become marginal. This change in habits has generated numerous consequences including the disfigurement of public space, the increase in land consumption, the increase in the emission of pollutants and the increase in the number of accidents. Moreover, new

forms of inequality and exclusion related to the different ability of populations to access places, use services, carry out activities and have relationships, exposing them to the risk of social exclusion, have recently arisen.

In fact, a city cannot be defined as a "Smart City" if it does not allow equal accessibility to everyone, including disabled individuals. As far as sustainable and inclusive accessibility to mobility is concerned, different comparisons between the entire network and the network suited to disabled people have been proposed to measure the equity/inequity of the network systems. According to Bartzokas-Tsiompras et al. [3], an observation of fifteen cities in Europe shows that an accessibility disparity between non-disabled and disabled (wheelchair user) individuals persists. In this direction, in addition to the policies to encourage active mobility, it is necessary to include strategies that guarantee equal accessibility (such as sidewalk presence and width, curb ramps, elevators, etc.) [4].

The communities and the national agenda are trying to implement innovative policies based on the use of renewable energy sources or that in any case limit the use of natural resources, favoring the growth of an economy oriented towards sustainable development. The challenge will be to improve the quality of life for citizens starting from urban mobility, which must become more sustainable by reducing the use of private vehicles (converting them to more sustainable technologies) and increasing active (pedestrian and cycle path) or public (trams, trains, buses, etc) forms of transport. This requires new multifunctional infrastructural networks capable of performing not only an ecological and environmental function but also an economic and social one. In relation to this, policies and methods aimed at fostering accessibility for disabled people may also play a critical role [3–5].

To achieve the climate change mitigation objectives, in line with the 2030 reference framework for climate and energy, the Italian Ministry of the Environment has launched and developed numerous actions that are aimed at promoting sustainable mobility systems. The main objectives are:

- promote smart mobility (car sharing, carpooling, smart parking and bike sharing);
- promote cycle and pedestrian mobility (paying attention to the inclusion of non-disabled and disabled individuals);
- discourage the circulation of private cars in downtown urban areas;
- promote local public transport.

At the base, however, there is a split between the needs of citizens, the responses of public actors, the transformations that have taken place in the social fabric and the demand for mobility. Precisely for this reason, extraordinary measures have been launched in recent years and important and massive resources have been identified in the sector.

With this funding, the necessary interventions (including communication and awareness campaigns) must be implemented to increase the safety of the movement of vulnerable users, thus encouraging active mobility that can be integrated with public transport. This is especially true for vehicles with restricted driving, such as trains, subways and modern trams; these forms of transport are able to rely on routes independent of ordinary traffic and offer attractive services in terms of travel times, but often do not offer much in terms of comfort. So, when long distances have to be covered, an intermodal exchange including active mobility can be an attractive alternative [6]. This becomes possible by making the transport and service system sustainable, integrated, accessible to all and interconnected, thanks also to the internet and information on mobility. The transport system (user/vehicle/road and environment) must therefore be reorganized in order promote cycle–pedestrian mobility by including it in transport plans and highlighting the health benefits (reduction in mortality and diseases such as hypertension, hypercholesterolemia, hyperglycemia, obesity, heart failure, etc. [7]).

According to numerous studies in the literature [8–10], in order to encourage citizens to travel on foot, it is necessary to plan and design pedestrian itineraries so that they are:

- safe: with reduced interactions with vehicles (parked or in motion);
- connected: so that they have continuity between the different origins and destinations;

- pleasant: well-sized and integrated into the territory, with slopes and lengths that are not excessive, etc.;
- usable for as many people as possible;
- accessible: with the removal of architectural barriers in order to connect the greatest number of environments (infrastructures, means of transport, etc.) present in the urban context and to guarantee equal accessibility to both non-disabled and disabled individuals [3–5].

This means that it is not enough to apply laws or regulations, it is necessary to analyze the environment as a whole according to a holistic approach, which includes a careful examination of the characteristics of the built environment to evaluate the mutual relationships of all the factors so that there is an integration between the man-made spaces and those who wish to use them.

Furthermore, it is worth mentioning that recent COVID−19 pandemic restrictions have highlighted the need to foster sustainable urban transport modes such as cycling and walking that can better mitigate the COVID−19 exposure risk that can be experienced on traditional public transport supply systems [11,12] In relation to this, the modelling of urban pedestrian activity can play a critical role in the planning of future smart cities.

Over the years, network analysis models have been developed [13,14] which have shown how the shape of a city influences the user's willingness to walk. To estimate how users are distributed on the different road sections, pure observational approaches associated with predictive models were used. The latter allowed predictions for the entire network despite the limited amount of data available. Among the most relevant are:

1. Sketch plan models: used as a planning guide as they attempt to approximate pedestrian demand based on a simple rule of thumb [15]. Pedestrian volumes are predicted through the use of counts and regression analyses as a function of land use (such as the surface of offices or commercial spaces) and/or travel generation indicators (parking capacity, transit volumes, movements of traffic, etc.) [16–20]. The advantage of these models is that they require minimal data collection and no knowledge of mathematical simulation or computer modeling. They are able to offer quick pedestrian volume estimates but are only effective at the aggregate level. These models have been applied in large urban environments where neither high precision nor detailed estimates are required [16,18,21].

2. Demand driven or origin–destination models. These models resemble traditional models of vehicular travel demand in many respects [22,23]; in fact, to estimate the pedestrian volume, they use four sub-models (trip generation and distribution, modal split and assignment). These approaches are based on the utility maximization theory, i.e., that all pedestrian actions are performed for a reason and therefore have utility with respect to a set goal. The disadvantage is linked to the fact that in the choice of the route a defined number of routes is considered [22,24] with few possible deviations. Ultimately, it can be said that these models offer complex descriptions of the built environment and the pedestrian behavior within it. However, they require a large volume of data, a high level of specialized technical skills and take time to set up, calibrate and review. To simulate and reconstruct pedestrian trips, two main modelling approaches can be usually identified.

   - Microsimulation models in which the behavior of the single user is reconstructed through his speed and interaction with other pedestrians, following predefined rules of behavior. Such models offer highly realistic simulations of small areas such as single streets or intersections and enclosed spaces such as transit centers, airports and shopping malls. This allows researchers to create large-scale forecasts of demand and travel volumes at a given point [25–28]. The most popular models are cellular automata [29,30], intelligent agents [31,32] and agent-based models [33,34]. These models allow a high level of detail to be obtained, but on the other hand, they require a good understanding of mathematics as well

as excellent computer skills. Furthermore, critical issues are related to their calibration based on experimental data.

- Macrosimulation models aggregate pedestrian movements in terms of flow, density and average travel speed. These are made up of partial differential equations from fluid dynamics; they take into account the conservation of mass and possibly an equilibrium equation of the moment [35]. The mathematical theory of models is based on temporal and one-dimensional dynamics, respecting the laws of conservation of flow. With macrosimulation models, it is possible to estimate demand, supply and their interaction, starting from statistical physical and behavioral laws (flow curves, generalized cost of transport, etc.).

3. Land use models (Seamless pedestrian models) are regression models used to explain "*the levels of demand recorded in the counts as a function of the measured characteristics of the adjacent environment*" [36]; in fact, they represent travel behavior based on the characteristics of an area (e.g., population density, employment density, family income, type of structure) and land use (schools, transit stops, parks, beaches, shops and civic facilities (libraries, post offices and government buildings)). They are simple to use as they take advantage of readily available data and methods [37]. Their drawback is that they are limited in terms of the acquisition of behavioral structure and are not transferable due to the relatively small sample size and characteristics upon which the models are built. In addition, the statistical significance is usually rather low.

4. Configurational models allow the ways in which the built environment can influence the dynamics of movement of pedestrians to be understood. These are more detailed than sketch plan models and can estimate volumes for street sections and intersections over an entire city or neighborhood. However, the models present varying techniques as a function of the amount of walking trips in a study area and the various algorithms for choosing the route. Most models consider the behavior of human beings; people do, in fact, choose linear paths that seem shorter to travel, as can be seen from the visual data [23,38]. Since these models are less complex than the others, they are also less expensive from a computational point of view; this allows for quick and easy modeling with different scenarios. This category includes the Space Syntax which uses a graphic "proximity" algorithm to estimate the movement potentials of pedestrians.

In this work, a configurational model was used to estimate the pedestrian flows (by carrying out simulations with the Space Syntax) which was improved by embedding numerical evaluations related to the different attractors present in the area. Numerous research has highlighted that pedestrian activity is linked to network connectivity, but at the same time it is influenced by other variables such as population, land use, the purpose of travel, means of connection, etc. [39–41].

The novel approach was based on a fusion of GIS-based software (namely QGIS) and Space Syntax analysis (by means of Toolkit) in order to provide a better estimate of pedestrian crowds. A flow-chart of the proposed approach is conveniently reported (see Figure 1).

The methodology and experimental calibration of this innovative modelling approach are detailed in the following section.

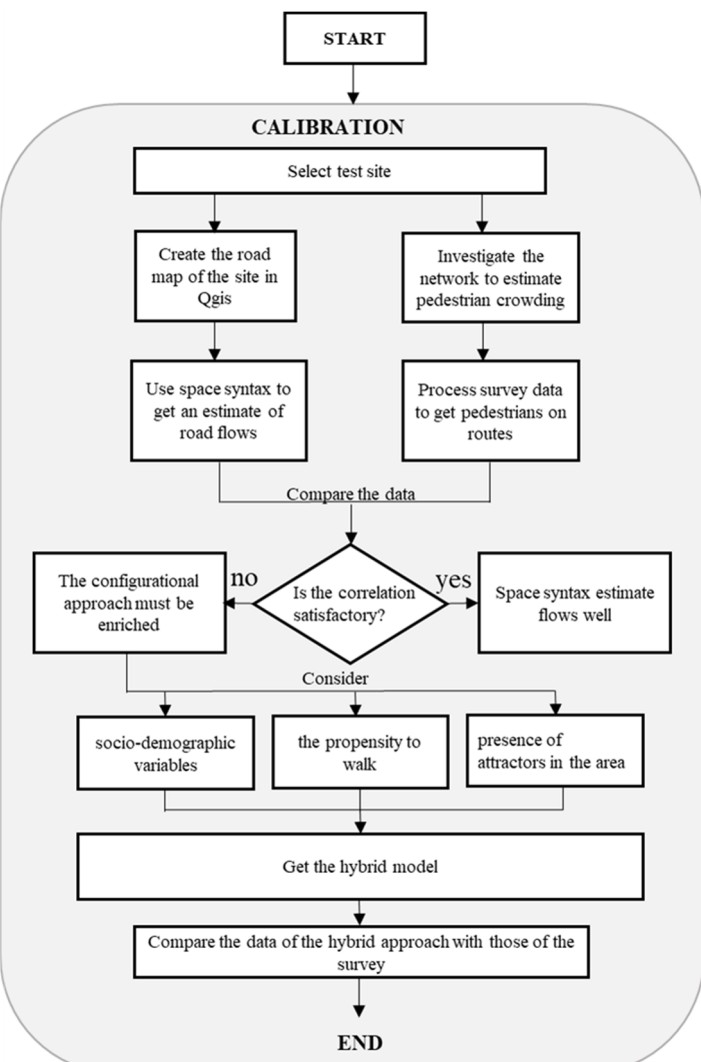

**Figure 1.** Flow chart of the proposed methodology.

## 2. Materials and Methods

Estimating crowding using the Space Syntax means carrying out a configurational analysis that is not related to the emission (o–d matrix) or to the modal choice but is attributable only to the shape of the network; this is the result of the history of urban evolution which has meant that city centers are made up of dense and connected road networks. In fact, the Space Syntax [42] consists of a series of theories and techniques for analyzing the spatial configuration of road networks and buildings created to help urban planners simulate the likely social effects generated by their projects.

According to this theory, there are some indicators that allow the configurational consistency of the elements to be described. Among these, the most used are:

- connectivity: it is a local variable that indicates the number of elements directly connected to a space;
- choice: represents the flow present through a space (a high value is obtained when there are short paths that connect several spaces);
- integration: it is a global measure of accessibility that indicates the minimum number of steps of a graph necessary to reach a point from another; this parameter can be evaluated either locally, by means of a so-called "topological radius" (usually equal to 3) or a buffer-based metric (usually 400 m) [43,44] which appears to be the best predictor of small-scale movements [45–47] or globally (with a topological radius n) which predicts large-scale movements (including vehicle movements, because people

on longer journeys will tend to read the grid in a more globalized way) [48]. This index presents several mathematical formulations [49–52], the most immediate seems to be the one reported by Raford and Regland [13]:

$$INT = \frac{2(MD - 1)}{K - 2} \tag{1}$$

where:

MD is the Mean Depth of the entire system; the Mean Depth is calculated by the equation:

$$MD = \frac{L}{(n - 1)} \tag{2}$$

where L is the Total Depth (that is the sum of all possible sections, given a starting point, to be traveled to reach a point of the network) and n is the total number of sections in the network;

K is the number of nodes in the network.

To obtain good results in terms of forecasting traffic flows, the most used analysis is the Angular Segment Analysis with Metric Radius; this fixes a buffer with a finite metric radius in which the connections between the sections are evaluated [53]. The radius used for the evaluation of pedestrian flows in an urban downtown is usually assumed to be 400 m, corresponding to a trip of about 5 min on foot. By starting the analysis, the configuration parameters of the network are obtained that best represent the probability of each section of being chosen or not in the various possible paths from a source to a destination.

Although the configurational approach has been proven to provide a fairly good estimate of pedestrian activity [14,45,54] it is believed that it cannot provide a reliable assessment on his own since population and land use data may help in improving the evaluation of pedestrian patterns throughout an urban network. Following this assumption, a hybrid approach was pursued that was able to take into account population and shopping activities. A description of this methodology is detailed in the following section

Before starting the analysis, it was necessary to reconstruct the road and pedestrian network on a GIS-based software (such as Qgis), creating the spatial relationship of contiguity between the census sections (Figure 2) and the road sections composing the road network. Each census section was represented with a centroid, each of which was attributed to the average value of the resident population (obtained from the Italian National Institute of Statistics (ISTAT) [55]) with the proximity approach.

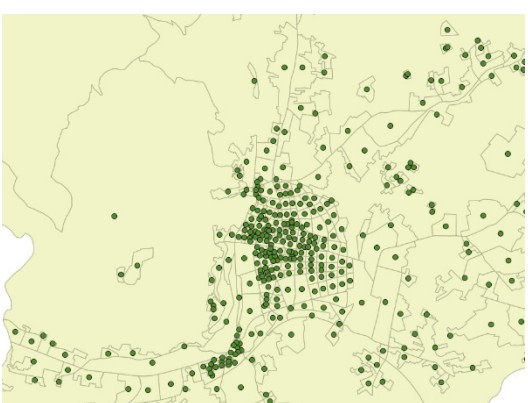

**Figure 2.** Example of area zoning into census sections (results derived by QGis).

A sequence of buffers with an increasing radius (*j*th) were considered, with radii from 100 to 1600 m (Figure 3), chosen on the basis of an impedance function (see Figure 4) [56].

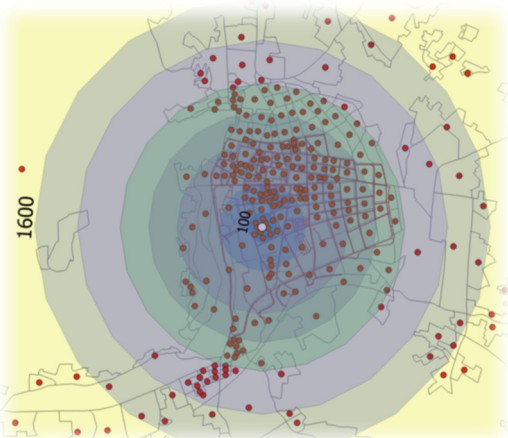

**Figure 3.** Example of an increasing radius buffer adopted in order to incorporate the gravitational approach in the configurational model. The main roads and pedestrian network were superimposed on the census block zoning (QGis results).

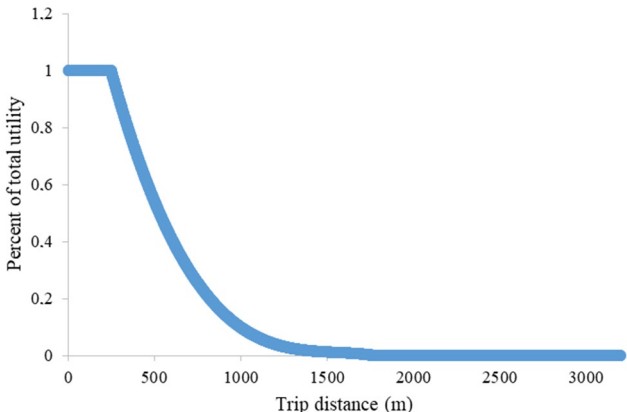

**Figure 4.** Pedestrian impedance function.

It describes the propensity to perform a trip (*k*) for pedestrians as the distance to travel increases.

By multiplying the average population values (*M*) of each circular crown by the impedance function, a coefficient can be obtained that expresses the ability of each section to attract pedestrian movements. By adding them it is possible to obtain a single (proximity weight) Prox weight for each centroid (*i*th).

$$P_{rox,i} = \sum_{j\,=\,100\text{m}}^{1600\text{m}} k_{Mj}\, M_{ij} \tag{3}$$

These weights were then used in the analysis with the Space Syntax and the results were compared with the data of a survey carried out on the network looking for a correlation [56].

However, it should be acknowledged that the population may not be able to capture the essential pedestrian activity that is much more affected by purposes other than those related to commuting. Based on these assumptions, a new modelling approach was investigated, taking into account the impact of the retail activities and other similar facilities and attractors on the overall pedestrian mobility.

Considering that the transport user moves towards these services on foot only if he is not forced to travel too much, the hybrid approach was "corrected" by incorporating an additional $K_{shop}$ factor.

This weight can be evaluated by considering not only the density of shops on each section of the network (expressed as the ratio between the number of shops present on the single section and the maximum number of shops evaluated on the entire network, in order to consider the attractiveness of the specific road section) but also the exposition length (expressed in terms of percentage of shops on the street front).

If there are no shops on the route, the gravitational criterion according to which the user tends to reach the adjacent road sections where a variety of services are provided was proposed. The gravitational weight $P_{(prox\_shop)}$ was estimated considering the influence of the surrounding area. According to the proposed approach, the number of shops falling within the circular crowns of 300 m and one kilometer was evaluated, these were multiplied by their attractiveness (i.e., by the respective impedance coefficient, see Figure 4) and normalized for the total number of shops in the area.

Summarizing, the $K_{shop}$ for a specific road section can be evaluated as:

$$K_{shop} = \begin{cases} P_{(prox\_shop)} * \frac{L_{mean\ shops}}{L_{total\ treats}} & \text{if } n^\circ{}_{shops} = 0 \\ P_{(prox\_shop)} * \frac{L_{mean\ shops}}{L_{total\ treats}} * \frac{n^\circ shops}{n^\circ \text{maximum number of stores on the network}} & \text{if } n^\circ{}_{shops} \neq 0 \end{cases} \quad (4)$$

where the $P_{(prox\_shop)}$ is evaluated as:

$$P_{(prox\_shop)} = \begin{cases} \frac{1*n^\circ{}_{shops\ 0–300\ m} + 0.3*n^\circ{}_{shops\ 300–1000\ m}}{n^\circ{}_{total\ shops\ 0–1000\ m}} & \text{if } n^\circ{}_{shops} = 0 \\ 1 & \text{if } n^\circ{}_{shops} \neq 0 \end{cases} \quad (5)$$

where n°shop 0–300 m represents the number of shops falling within the 300 m radius buffer and n°shop 300–100 m is the number of shops falling within the buffer comprised between the 1000 m and the 300 m radius.

The new $K_{shop}$, factor can be incorporated in the evaluation of weight following the configurational analysis in order to assess the correlation with pedestrian activity for each examined road section. The methodology was applied to a case study in a typical town of central Italy. Pedestrian counts were collected by means of an original measuring procedure and the calibration of the model was obtained that is detailed in the following section.

## 3. Results

The study area on which the approach was calibrated was the downtown area of Cassino, an Italian town (Figure 5) of about 36,000 inhabitants in the province of Frosinone in Lazio.

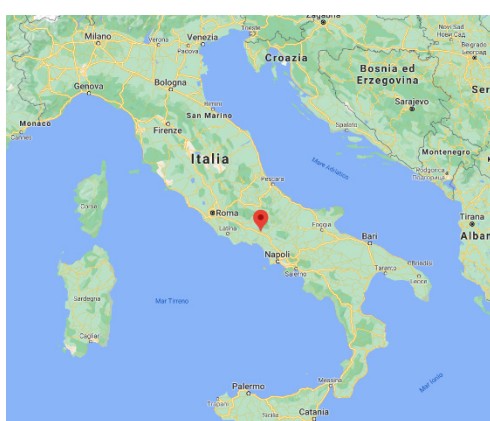

**Figure 5.** Location of Cassino in Italy.

The city of Cassino (Figure 6) was chosen as the case study for the layout of its network: relatively small-size, easy accessibility, the presence of transport infrastructures (train station, bus services, bike sharing service, etc.), walkability attitude (mainly flat territory, etc.). Furthermore, the presence of schools, a university, restaurants, shops, public

and private offices, etc. guaranteed a regular movement of people for ordinary daily activities. These features made the model easy to replicate without strong limitations or manipulations.

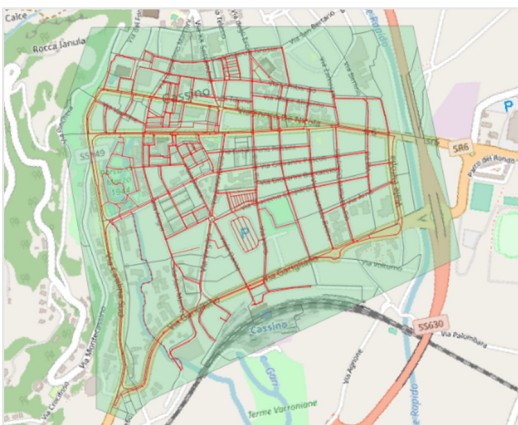

**Figure 6.** The city center of Cassino.

The study area was discretized into census sections that were each represented by a centroid. Once the population information was transferred from the section centroids to the network centroids, the respective values of the proximity weights (Prox) were associated with the network segments and then the hybrid forecast analysis was started. It has been observed that the pure configurational approach does not seem to fully capture urban pedestrian activities, so a hybrid approach was proposed that integrated the configurational analysis obtained by the Space Syntax with land use and pedestrian travel behavior.

The results, differentiated according to the type of weight used, are shown on the screen with the respective chromatic representations of the net.

It can be observed that the configurational analysis (Figure 7), which refers only to the geometric and topological characteristics of the network, showed a greater connection between the sections of the center area, in which the highest integration values were achieved. As can be seen, the integration values decreased moving away from the Central Business District (CBD) towards the peripheral areas with less connectivity.

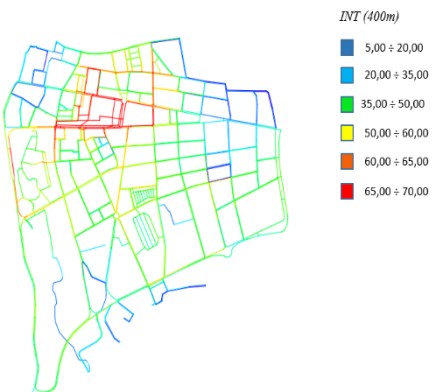

**Figure 7.** Result of the pure configurational analysis provided by the Space Syntax Toolkit.

In the hybrid analysis (Figure 8), the combined effect of the geographical distribution of the population and the pedestrian willing to move was considered. It was observed that the central area continued to have high integration values, due to the connectivity of the sections, however high values were also observed in the lower area (near the railway station) and at the eastern edge of the study area. In fact, in these areas, there is a greater resident population, which generates a more intense number of daily movements.

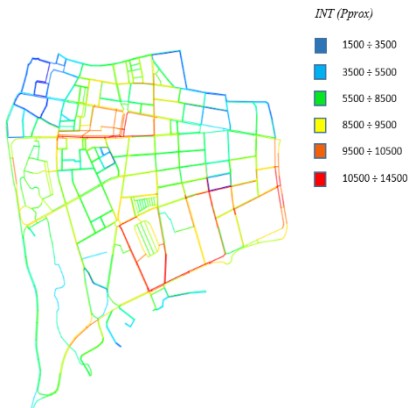

**Figure 8.** Hybrid analysis result provided by the Space Syntax Toolkit.

The data obtained by the Space Syntax, both with the pure configurational analysis and with the hybrid analysis, needed to be compared with the on-site pedestrian counts in order to evaluate how much the model was able to represent reality. Therefore, it was necessary to carry out surveys on the network to estimate pedestrian crowding. Over the years, numerous monitoring techniques have been developed to identify pedestrian flows, which can be divided into two types: automatic and manual detection. The difference between the two lies in the workforce deployed. The first uses instruments with high precision and sensitivity (such as pressure sensors, infrared devices and digital instruments capable of recording and storing video and position data capable of interacting via the Internet with other computer systems) and are therefore very expensive. The second consists of the acquisition of data based on the direct survey of an operator; it is cheaper in terms of costs but on the contrary it is more time-consuming.

In this work, an innovative approach based on a "moving observer" concept was used. This practice, usually adopted to evaluate the main parameters of the vehicular flow, is based on a series of surveys carried out by two or three individuals operating onboard an observation vehicle, which covers the route to be analyzed in both directions of travel and at a constant and reduced speed (less than 30 km/h).

This technique, however, involves the use of multiple operators to fill in special forms created during the survey design phase. To overcome this problem and to allow a single operator to carry out the survey, the vehicle was equipped with technological supports for video shooting. The system conceived allowed the continuous capture and storage of data, showing the date, time and GPS location.

To take into account the unpredictability and variability of human behavior (a function of the urban environment and its attractive capabilities) it was necessary to take into account the time slots and critical days. This made it possible to define two itineraries (one for a weekday and one for a holiday) (Figures 9 and 10), both with the same start and endpoint. The main network was broken down into a series of road arcs ending in the nodes identified by the intersections, with and without traffic lights, numbered in progressive order.

In a subsequent phase, the data obtained from the surveys were processed by an operator who, after viewing the videos, reported on a spreadsheet the number of pedestrians in each section and their relative position (stopped, moving or crossing).

Following the calibration of the different proposed models, from the comparison of the survey data with the results of the Space Syntax it was possible to observe that there were singular points dispersed therefore the trend line presented, in all cases, a very low coefficient of determination ($R^2$) (Figure 11).

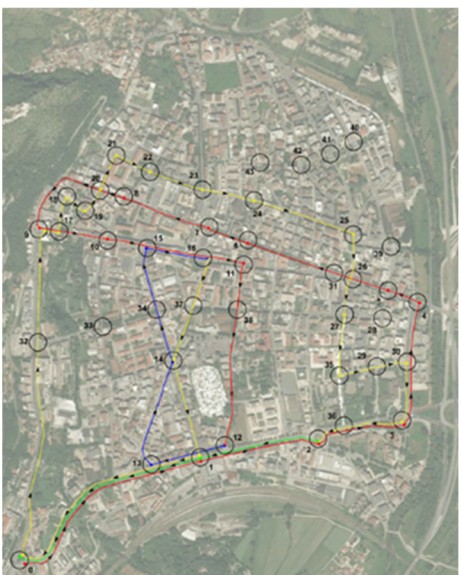

**Figure 9.** Weekday route.

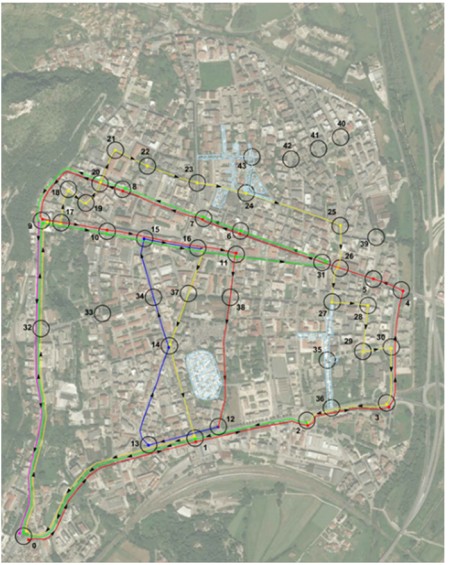

**Figure 10.** Route of the days before holidays.

A careful analysis of the area was carried out. It was pointed out that these points represented the areas of greatest attraction grouped under the name Shops (ie bars, restaurants, groceries, shoe shops, clothing stores, etc.) (Figure 12).

An additional survey on retail activities was carried out on Saturdays when several open market areas were operating in the town center of Cassino, thus providing a dramatic re-evaluation of the $K_{shop}$ values for most of the examined road sections.

Collected pedestrian counts were compared with INT(Prox)*$Ks_{hop}$ values for all the investigated road sections. The results are conveniently reported in Figure 13 and a preliminary calibration with a statistical linear regression model has also been depicted for weekday and pre-holidays.

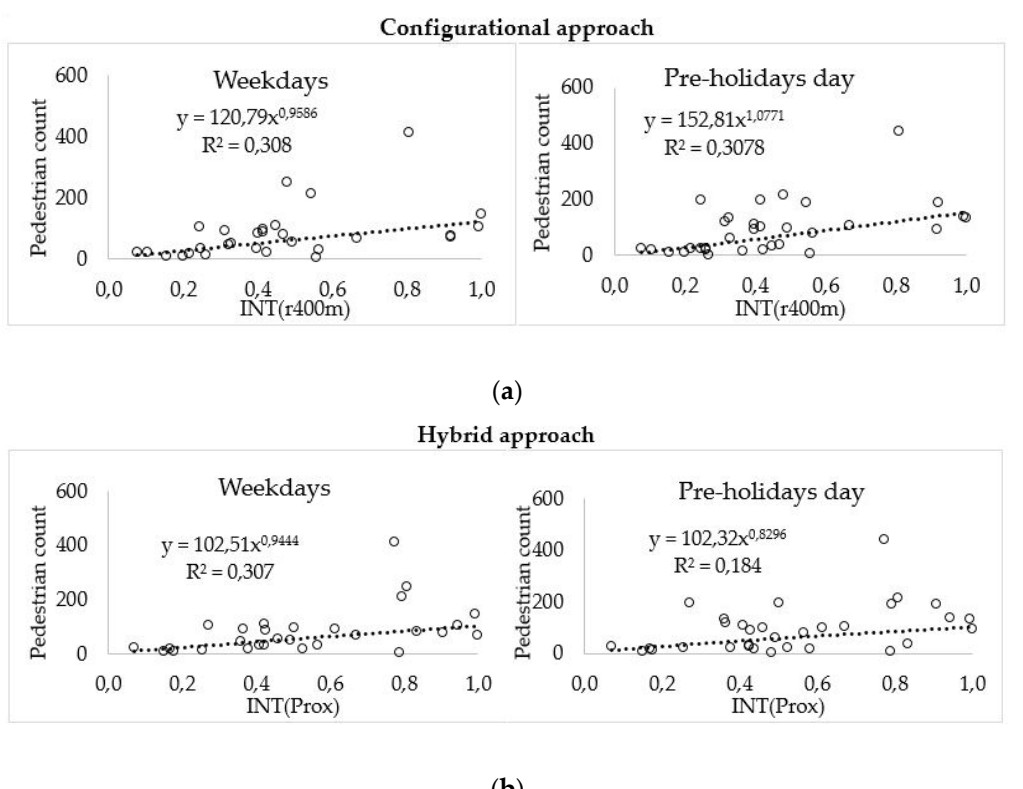

**(a)**

**(b)**

**Figure 11.** Comparison between the findings and the results of the Space Syntax: (**a**) Configurational approach and (**b**) Hybrid approach.

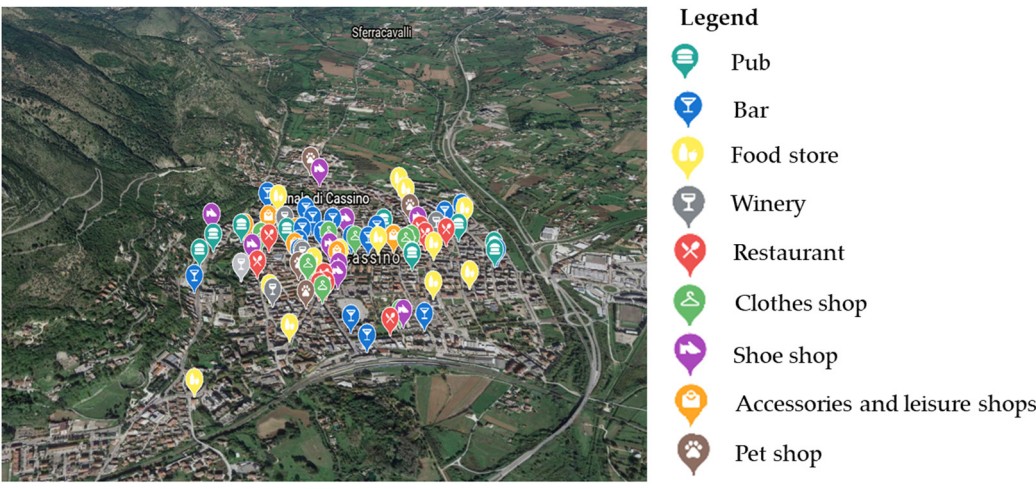

**Figure 12.** Shops in the city center.

The results (Figure 13) show that the hybrid model was better able to estimate the pedestrians present on the different road sections; in fact, in the preliminary calibration with a statistical linear regression model, a good correlation was observed.

The corresponding regression statistics of the obtained model were also conveniently reported. An overall comparison between the estimated pedestrian counts and the measured ones is also depicted in Figure 13. A Pearson coefficient of correlation, P, of nearly 0,9 was also derived. For a better interpretation of the final regression model, a statistical analysis was performed and the results are reported in Table 1.

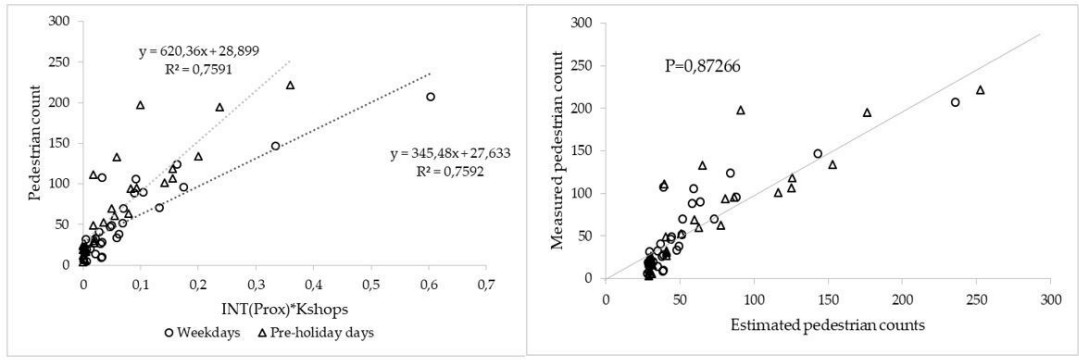

**Figure 13.** Results with weighted hybrid analysis.

**Table 1.** Results of statistical analysis.

| | Pre-Holiday Days | Weekdays |
|---|---|---|
| Model | $Y = ax + b$ | $Y = ax + b$ |
| N° observations | 32 | 29 |
| Sum of residuals | −8.05407296411431E-10 | −6.51939657814182E-9 |
| Average residual | −2.51689780128572E-11 | −2.24806778556615E-10 |
| Residual sum of squares (Absolute) | 27.216.1823810779 | 15.919.5603658666 |
| Residual sum of squares (Relative) | 27.216.1823810779 | 15.919.5603658666 |
| Standard Error of the Estimate | 30.119861875003 | 24.2819551701248 |
| Coeff. of Multiple Determination ($R^2$) | 0.7590762769 | 0.7592199078 |
| Proportion of Variance Explained | 75.90762769 % | 75.92199078 % |
| Adjusted coeff. of Multiple Determination ($Ra^2$) | 0.7510454862 | 0.7503021266 |
| Durbin–Watson statistic | 1.41447158255192 | 1.8936011290373 |

| | | (a) Regression Variable Results | | | (b) Regression Variable Results | | | |
|---|---|---|---|---|---|---|---|---|
| Variable | Value | St. Er. | t-ratio | Prob(t) | Value | St. Er. | t-ratio | Prob(t) |
| a | 620.3528 | 63.8080 | 9.7221 | 0.00000 | 345.4754 | 37.4422 | 9.2268 | 0.00000 |
| b | 28.8985 | 6.5563 | 4.4077 | 0.00012 | 27.6331 | 5.4085 | 5.1091 | 0.00002 |

## 4. Discussion

Pedestrian activity and patterns are dramatically affected by variability in space and time. The uncertainty level for this transport mode is undoubtedly higher than that experienced for vehicular traffic that, in turn, is more related to commuter behavior. Within this context, a configurational analysis has been effectively used in the past, provided that pedestrian counts on a daily basis (often collected by means of video recordings provided by urban surveillance systems) were made available.

However, data collection via those traditional approaches may be extremely cumbersome and time-consuming. A "moving observer" approach may be more effective in collecting relevant data on pedestrians. On the other hand, pedestrian counts collected by means of this latter method are affected by higher dispersion since data are usually averaged on a smaller time span. As a matter of fact, a comparison with estimates provided by the traditional configurational approach yielded fairly unsatisfactory results in terms of the correlation coefficient (see Figure 11a).

However, it was observed that the incorporation of a "surrogate" gravitational approach by means of an impedance function able to capture the pedestrian propensity to perform a trip on foot as a function of traveled distance, did not add much value to the forecasting performance of the proposed approach (see Figure 11b).

Following these results, an in-depth study on retail activities was carried out on both weekday and pre-holiday days since several open markets are set up on Saturdays.

The idea behind the methodology was to exploit the additional data related to secondary (not commuter) activities that can be induced by retail attractors. The Cassino area offered a "unique" opportunity to test the validity of this assumption as on Saturdays

several provisional open markets are hosted in the city of Cassino, therefore it was possible to discriminate the different "density" of retail activities to the extent that they may affect pedestrian counts.

It should be noted that the results with this "enriched" hybrid approach, via the INT(Prox)*$K_{shop}$ value, provided a fairly good agreement with the collected pedestrian data in both analyzed scenarios. A statistical linear regression model was used in order to describe the correlation between the measured pedestrian count per single passage and the independent variable, INT(Prox)*$K_{shop}$.

It is worth noting that the regression models other than the linear one did not show a similar significant agreement (correlation) with the measured data. As a matter of fact, despite its naïve nature, the linear model seemed to highlight some important results for the two examined scenarios (weekdays and pre-holidays):

- the constant term appears to be similar for both weekdays and pre-holiday days, thus providing evidence that there is a "basic level" of pedestrian activity that is irrespective of the specific day of week;
- the slope of the model was apparently different; as a matter of fact, it was clearly higher for pre-holiday days, thus providing evidence that there is a greater propensity of citizens to move on foot in the days before holidays (perhaps also due to the presence of several local market areas that are able to attract more customers also coming from the surrounding towns).

However, by comparing the data (Figure 13), it can be also seen that, with the Pearson coefficient close to unity, the quantitative variables used were well correlated and the model was statistically significant.

## 5. Conclusions

The fight against pollution has made sustainability a global problem, especially for cities and their development. The first goal to make cities greener is to promote sustainable mobility, thus favoring traditional modes of travel that have the advantage of being completely ecological with zero emissions. From this perspective, the correct and efficient planning of the infrastructural network is necessary to guarantee safety, inclusivity and comfort for users. To ensure that soft mobility will spread, as it did for road infrastructures dedicated to vehicles, cycle and pedestrian paths must also be included in programmatic projects and respect some specific characteristics. First of all, it is useful to have an overview of the territory and the need to move people, making sure that a network of connections that uniformly covers the area and is comfortable is created. In designing these paths, in order to size them correctly, it is necessary to analyze the flows of people moving into the study area and to define the main paths, the origin, the direction and the purposes with which certain journeys are made. To determine the flows in the past it was thought that it was sufficient to know the configuration of the network, with the belief that the movements took place only in the vicinity of the residence. Actually, in modern cities, where services are scattered throughout the territory, it is necessary to think of approaches that make it possible to merge the purely configurational aspects with the characteristics of the territory that attracts users. In this work, a hybrid approach was developed which has provided a satisfactory estimate of pedestrian movements and which can be a valid support for the study of accidents related to vulnerable users in the urban area.

Furthermore, it has to be underlined that the ability to capture the complex pedestrian activity may represent an effective tool in evaluating the risk exposure to pandemics or security (terroristic attacks) threats since it seems acceptable that urban areas are by far the most vulnerable locations with reference to this kind of hazard.

Preliminary results following an experimental campaign carried out by means of an innovative "moving observer" approach seem very promising. However, it has to be acknowledged that the proposed methodology deserves further experimental validation with additional pedestrian data. It is also planned to extend the use of the proposed methodology to other urban contexts.



Surely this methodology will help town managers to evaluate the risks from the pandemic as well as security or road safety risk exposure of pedestrian flows. The knowledge of pedestrian crowd levels may help mitigation actions such as accessibility limitations/restrictions to be undertaken in the case of critical events.

Furthermore, as far as road safety is concerned, provided that pedestrian activities estimates can be integrated with traffic volumes and motor vehicle speeds, it is believed that this approach may allow planners to take a proactive role in risk assessment (linked to pedestrian exposure) when no data are available for pedestrian–car collisions.

**Author Contributions:** Conceptualization, D.S., M.D. and V.N.; methodology, D.S., M.D. and V.N.; validation, D.S. and A.E.; formal analysis, D.S. and A.E.; investigation, D.S. and A.E.; resources, D.S.; A.E.; M.D. and V.N.; data curation, D.S. and A.E.; writing—original draft preparation, D.S. and M.D.; writing—review and editing, A.E. and V.N.; supervision, M.D. and V.N.; All authors have read and agreed to the published version of the manuscript.

**Funding:** This research received no external funding.

**Institutional Review Board Statement:** Not applicable.

**Informed Consent Statement:** Not applicable.

**Data Availability Statement:** Data sharing not applicable.

**Conflicts of Interest:** The authors declare no conflict of interest.

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
