# Peer review of "Towards Sustainability: New Tools for Planning Urban Pedestrian Mobility"

_sustainability, doi:10.3390/su13169371_

Round 1
Reviewer 1 Report
Interesting approach.
Clearly Specify the methods involved and justification of the development of the final regression model that predicts pedestrian counts. Can you claim a cause and effect relationship or is it just prediction?
Better interpretation of the final regression model and discussion on it usefulness is necessary.
Some further statistical analysis is necessary to provide evidence for the validity of the model.
Check for typographical/grammar errors.
Author Response
Recensione 1:
Richieste modifiche moderate dell'inglese.
Risposta: Ogni volta che vengono rilevati, gli errori tipografici/grammaticali sono stati opportunamente corretti.
Commenti e suggerimenti per gli autori
Approccio interessante.
Specificare chiaramente i metodi coinvolti e la giustificazione dello sviluppo del modello di regressione finale che prevede il conteggio dei pedoni. Puoi rivendicare una relazione di causa ed effetto o è solo una previsione?
Risposta: In effetti, come chiaramente affermato nella sezione discussione e conclusione, l'approccio configurazionale puro (Space Syntax) sembra fornire una stima scadente del conteggio dei pedoni quando i dati vengono raccolti da un approccio di osservatore in movimento. L'idea alla base della metodologia è quella di sfruttare i dati aggiuntivi relativi alle attività secondarie (non pendolari) che possono essere indotte dagli attrattori al dettaglio. La zona di Cassino ha offerto un'occasione “unica” per testare la validità di tale presupposto in quanto il sabato sono ospitati nella città di Cassino diversi mercati aperti provvisori, pertanto è stato possibile discriminare diverse “densità” di attività di vendita al dettaglio nella misura in cui possono incidere conta pedoni. A questo proposito, può essere invocato un rapporto di causa ed effetto. Per spiegare meglio il metodo proposto,
È necessaria una migliore interpretazione del modello di regressione finale e una discussione sulla sua utilità.
Risposta: Come accennato nel paragrafo "Discussione", il modello lineare permette di evidenziare che, nei due scenari in esame (feriali e prefestivi), nell'area di studio è presente un'attività pedonale di base (legata alla nota termine della retta) mentre il diverso coefficiente angolare mostra una diversa propensione a viaggiare (maggiore nei giorni prefestivi rispetto ai giorni feriali). Per migliorare la chiarezza, è stata aggiunta una migliore interpretazione del modello di regressione finale e una discussione sulla sua utilità nella sezione dei risultati e della discussione e conclusione.
Sono necessarie ulteriori analisi statistiche per fornire prove della validità del modello .
Risposta: Sono state aggiunte ulteriori analisi statistiche (vedi paragrafo “Risultati”).
Verifica la presenza di errori tipografici/grammaticali.
Risposta: Ogni volta che vengono rilevati, gli errori tipografici/grammaticali sono stati opportunamente corretti.

Reviewer 2 Report
The manuscript has some grammatical errors.
It is appropriate to include some sentences related to the current pandemic and the spread of smart cities and sustainable mobility.
It is recommended to insert a flow chart of the research steps conducted at the end of the introduction
I also recommend reading the following works
1)Shorfuzzaman, M., Hossain, M. S., & Alhamid, M. F. (2021). Towards the sustainable development of smart cities through mass video surveillance: A response to the COVID-19 pandemic. Sustainable cities and society, 64, 102582.
2) Kunzmann, K. R. (2020). Smart cities after covid-19: ten narratives. disP-The Planning Review, 56(2), 20-31.
The concept of cities with accessibility for all should be emphasized. We recommend reading the following works
1) Bartzokas-Tsiompras, A., Paraskevopoulos, Y., Sfakaki, A., & Photis, Y. N. (2020, June). Addressing street network accessibility inequities for wheelchair users in fifteen European city centers. In Conference on Sustainable Urban Mobility (pp. 1022-1031). Springer, Cham.
2)Campisi, T., Ignaccolo, M., Inturri, G., Tesoriere, G., & Torrisi, V. (2020). Evaluation of walkability and mobility requirements of visually impaired people in urban spaces. Research in Transportation Business & Management, 100592.
3) Campisi, T., Basbas, S., Tesoriere, G., Trouva, M., Papas, T., & Mrak, I. (2020). How to Create Walking Friendly Cities. A Multi-Criteria Analysis of the Central Open Market Area of Rijeka. Sustainability, 12(22), 9470.
Figures 1-2 and 6-7 should be better argued and the source included.
Justify the choice of case study and describe whether the methodology is replicable and whether the study has any limitations.
A legend with the different symbols should be added to Figure 11.
Figure 12 should be further explained
Author Response
Review 2:
English language and style are fine/minor spell check required.
Answer: Whenever detected, typographical/grammar errors have been conveniently corrected.
Comments and Suggestions for Authors
The manuscript has some grammatical errors.
Answer: Whenever detected, spell check errors have been conveniently corrected.
It is appropriate to include some sentences related to the current pandemic and the spread of smart cities and sustainable mobility.
I also recommend reading the following works
1) Shorfuzzaman, M., Hossain, M. S., & Alhamid, M. F. (2021). Towards the sustainable development of smart cities through mass video surveillance: A response to the COVID-19 pandemic. Sustainable cities and society, 64, 102582.
2) Kunzmann, K. R. (2020). Smart cities after covid-19: ten narratives. disP-The Planning Review, 56(2), 20-31.
The concept of cities with accessibility for all should be emphasized. We recommend reading the following works
1) Bartzokas-Tsiompras, A., Paraskevopoulos, Y., Sfakaki, A., & Photis, Y. N. (2020, June). Addressing street network accessibility inequities for wheelchair users in fifteen European city centers. In Conference on Sustainable Urban Mobility (pp. 1022-1031). Springer, Cham.
2) Campisi, T., Ignaccolo, M., Inturri, G., Tesoriere, G., & Torrisi, V. (2020). Evaluation of walkability and mobility requirements of visually impaired people in urban spaces. Research in Transportation Business & Management, 100592.
3) Campisi, T., Basbas, S., Tesoriere, G., Trouva, M., Papas, T., & Mrak, I. (2020). How to Create Walking Friendly Cities. A Multi-Criteria Analysis of the Central Open Market Area of Rijeka. Sustainability, 12(22), 9470.
Answer: The authors would like to thank the reviewer for the suggested works which are very interesting for our activities. According to the pertinent observations, the text has been improved including pandemic and accessibility aspects. The suggested works have been carefully examined and mentioned in the paper.
It is recommended to insert a flow chart of the research steps conducted at the end of the introduction.
Answer: A flow chart has been inserted to describe the proposed methodology (see new Figure 1)
Figures 1-2 and 6-7 should be better argued and the source included.
Answer: Figures 1-2 and 6-7 (now corresponding to 2-3 and 7-8) have been have been better argued and the source has been included.
Justify the choice of case study and describe whether the methodology is replicable and whether the study has any limitations.
Answer: Case study has been conveniently justified in the discussion section and replicability and limitations of methodology have been reported in the conclusion section.
A legend with the different symbols should be added to Figure 11.
Answer: In the figure 11 (now figure 12), a legend with the different symbols has been added.
Figure 12 should be further explained
Answer: Figure 12 (now 13) has been further explained in the text.
